# Conserving Potential and Endangered Species of *Pericopsis mooniana* Thwaites in Indonesia

**Julianus Kinho** [1,*], **Suhartati** [2], **Husna** [3], **Faisal Danu Tuheteru** [3], **Diah Irawati Dwi Arini** [1], **Moh. Andika Lawasi** [4], **Resti Ura'** [1], **Retno Prayudyaningsih** [5], **Yulianti** [2], **Subarudi** [6], **Lutfy Abdulah** [1], **Ruliyana Susanti** [1], **Totok Kartono Waluyo** [7], **Sona Suhartana** [7], **Andianto** [8], **Marfuah Wardani** [2], **Titi Kalima** [1], **Elis Tambaru** [9], **Wahyudi Isnan** [1], **Adi Susilo** [1], **Ngatiman** [10], **Laode Alhamd** [1], **Dulsalam** [7] and **Soenarno** [7]

[1] Research Center for Ecology and Ethnobiology, National Research and Innovation Agency, Jalan Raya Jakarta-Bogor Km 46, Cibinong 16911, Indonesia

[2] Research Center for Plant Conservation, Botanic Gardens, and Forestry, National Research and Innovation Agency, Jalan Raya Jakarta-Bogor Km 46, Cibinong 16911, Indonesia

[3] Department of Forestry, Faculty of Forestry and Environment Science, Halu Oleo University, Jl. HEA Mokodompit, Kendari 93121, Indonesia

[4] Research Center for Society and Culture, National Research and Innovation Agency, Jl. Gatot Subroto No. 10, Mampang Prapatan, Jakarta 12710, Indonesia

[5] Research Centre for Applied Microbiology, National Research and Innovation Agency, Jalan Raya Jakarta-Bogor Km 46, Cibinong 16911, Indonesia

[6] Research Center for Population, National Research and Innovation Agency, JL. Gatot Subroto No. 10, Mampang Prapatan, Jakarta 12710, Indonesia

[7] Research Center for Biomass and Bioproducts, National Research and Innovation, Jalan Raya Jakarta-Bogor Km 46, Cibinong 16911, Indonesia

[8] Research Center for Biosystematics and Evolution, National Research and Innovation Agency, Jalan Raya Jakarta-Bogor Km 46, Cibinong 16911, Indonesia

[9] Department of Biology, Faculty of Mathematics and Natural Sciences, Hasanuddin University, Jl. Perintis Kemerdekaan Km 10, Makassar 90245, Indonesia

[10] Research Center for Applied Zoology, National Research and Innovation Agency, Jalan Raya Jakarta-Bogor Km 46, Cibinong 16911, Indonesia

* Correspondence: julianus.kinho@brin.go.id

**Abstract:** Indonesia has around 4000 wood species, and 10% (400) of species are categorized as commercial wood. One species is kayu kuku (*Pericopsis mooniana* Thwaites), native to Southeast Sulawesi. This species is considered a fancy wood used for sawn timber, veneer, plywood, carving, and furniture. The high demand for wood caused excessive logging and threatened its sustainability. In addition, planting *P. mooniana* has presented several challenges, including seedling production, viability and germination rate, nursery technology, and silviculture techniques. As a result, the genera of *Pericopsis*, including *P. elata* (Europe), *P. mooniana* (Sri Lanka), and *P. angolenses* (Africa), have been listed in the Convention on International Trade in Endangered Species (CITES) Appendix. Based on The International Union for Conservation of Nature (IUCN) Red List of Threatened Species, *P. mooniana* is categorized as Vulnerable (A1cd). This conservation status has raised issues regarding its biodiversity, conservation, and sustainability in the near future. This paper aims to review the conservation of potential and endangered species of *P. mooniana* and highlight some efforts for its species conservation and sustainable use in Indonesia. The method used is a systematic literature review based on *P. mooniana's* publication derived from various reputable journal sources and additional literature sources. The results revealed that the future demand for *P. mooniana* still increases significantly due to its excellent wood characteristics. This high demand should be balanced with both silviculture techniques and conservation efforts. The silviculture of *P. mooniana* has been improved through seed storage technology, improved viability and germination rates, proper micro and macro propagation, applying hormones, in vitro seed storage, improved nursery technology, and harvesting techniques. *P. mooniana* conservation can be conducted with both in situ and ex situ conservation efforts. In situ conservation is carried out by protecting its mother trees in natural conditions (i.e., Lamedae Nature Reserve) for producing good quality seeds and seedlings. Ex situ conservation is realized by planting seeds and seedlings to produce more wood through rehabilitating

and restoring critical forests and lands due to its ability to adapt to marginal land and mitigate climate change. Other actions required for supporting ex situ conservation are preventing illegal logging, regeneration, conservation education, reforestation, agroforestry system applied in private and community lands, and industrial forest plantations.

**Keywords:** fancy wood; economic value; sustainable management; conservation

## 1. Introduction

Indonesia has around 4000 wood species that can be used for building construction. However, only 10% (400) of wood species have economic value, and 260 species are commercially traded in national and international markets [1]. One of the commercial and tropical wood species is kayu kuku (*Pericopsis mooniana* Thwaites.), a native species in Southeast Sulawesi [2]. This species has good characteristics such as being a luxurious, fancy, and expensive wood with high economic value in the international market [2,3]. Its wood can be used as household tools and veneer and is suitable for heavy construction such as ship decks, bridges, railway wood sleepers, frame wood, and truck tailgate [4].

Another "tropical wood" species is *P. elata* (Harms) Meeuwen., known as African satinwood and gold teak [5]. This species is quite famous for its timber due to its similarity of texture and other wood characteristics to those of teak (*Tectona grandis* L.). This timber is traded internationally for exterior and interior work [5]. Like *P. elata* timber in Africa, *P. mooniana* timber from Indonesia has been traded and utilized increasingly without replanting efforts [3], exported since 1972, and exploited uncontrollably [4]. Consequently, the species' population is decreasing, as Rain Forest Action reported in 2004 [6]. Due to its high demand and excessive harvesting in the international market, trees from the genera *Pericopsis* have been included on the IUCN red list. For example, *P. elata* is listed as Endangered [5,7], *P. mooniana* as Vulnerable [8,9], and *P. laxiflora* (Benth. ex Baker) Meeuwen. as the Least Concern species [10].

The *Pericopsis* trees grow naturally in the Lamedae forest, including the natural reserve forest, Kolaka district, and Southeast Sulawesi [9,11]. Generally, the trees are found in beach forests, near the river bank, and in forests with an elevation of 200–350 m asl [3]. Nkulu et al. [12] found *P. angolensis* (Baker) Meeuwen. dominant in a habitat based on the Mg concentration above 140 $\mu g.g^{-1}$ and soil Al concentration at a threshold value of 220 $\mu g.g^{-1}$. The natural distribution of Afrormosia (*P. elata*) ranges from Ivory Coast, Ghana, Cameroon, DRC, Congo-Brazzaville, CAR, and Nigeria [5]. *P. mooniana* is a native species found in wet zone floodplain forests in Sri Lanka [13].

Indonesia owns 120 million hectares of forest, accounting for 64% of the country's total land area [14], including 14 million ha of critical forests and land. Meanwhile, *P. mooniana* is well suited for rehabilitation, particularly for ex-mining reclamation land [15]. This tree is a pioneer species due to its adaptability to marginal and unproductive lands [16,17]. This has been supported by research findings of Perala and Wulandari [18] that the tree has a high survival rate of 63% in gold-tailing lands, reaching 92% with additional treatment of vermicompost, rhizobium, and mycorrhiza. This indicated that the trees could grow and develop well in all forest areas, implying the potency of this species will become a mainstay and superior tree in the near future in Indonesia.

*P. mooniana* seed germination is only 68% [19], and its seedlings have been predicted to be well growing (100%) with various land media, tailings, and combinations between tailing and manure [16]; however, it is still questionable regarding its growth after planting in various types of forest lands. Therefore, research on seed production and the silviculture techniques of *P. mooniana* is needed to optimize its growth and species conservation in response to its high use and demand in the future. This paper aims to review the conserving potential and endangered species of *P. mooniana* and highlight some efforts for this species' conservation and sustainable use in Indonesia.

## 2. Methods

The method used is the systematic literature review. The review was conducted based on publications of *P. mooniana* derived from various reputable sources and additional literature sources to capture the state of conserving this species as potential and endangered species. Some keywords in English and Indonesian were used to find relevant issues by employing a search engine. An intensive search for online publications for 1990–2022 was carried out in August–September 2022. Literature references were collected through search with the Google search engine, Google Scholar, Science Direct, ResearchGate, Crossref, Scopus, PubMed, and other relevant databases. The stages of searching and screening the publications are shown in Figure 1.

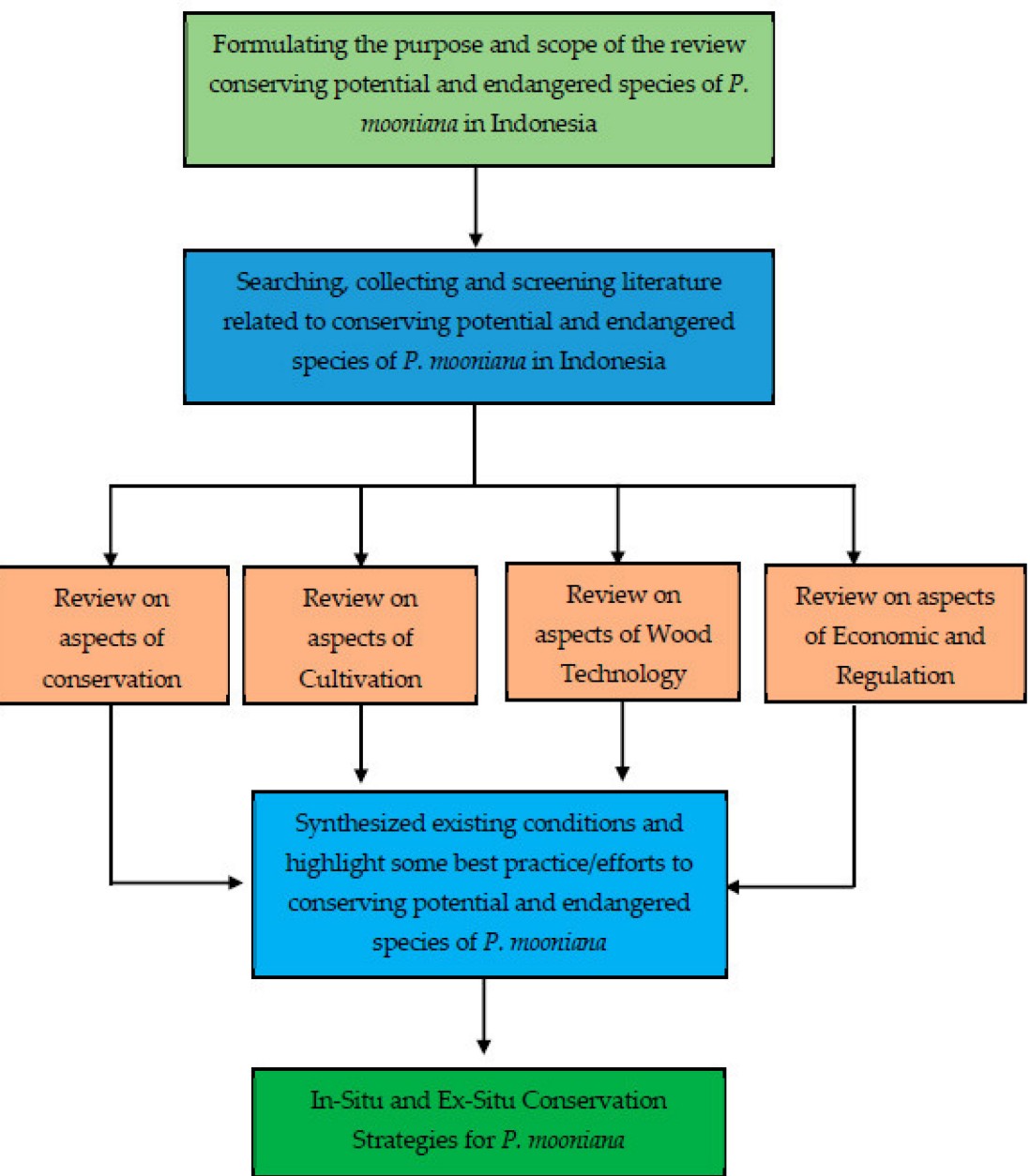

**Figure 1.** Stages in conducting the review.

## 3. Results

### 3.1. Conservation

3.1.1. Botanical Description of Kayu Kuku (*P. mooniana*)

Classification of kayu kuku (*P. mooniana*) [20]:

Kingdom: Plantae
Division: Spermatophytae
Sub division: Angiospermae
Class: Magnoliopsida
Order: Fabales
Family: Fabaceae
Genus: *Pericopsis*
Species: *Pericopsis mooniana* Thwaites.

*P. mooniana* has a tree habitus and medium size. In natural forests, the trunk height reaches 30–40 m; the branch-free trunk (clear bole) can reach 3/4 part of its total height, and the diameter reaches 35–100 cm [21]. The main stem is straight, shallowly grooved, and notched at the base; the stem is thin and smooth reddish, and the leaves are crossed opposite. The bark is light brown. This species has brighter colored sapwood than its reddish-brown heartwood [6] and belongs to the I-II durable class wood type [22]. The wooden surface is slippery and shiny and has decorative lines [23]. *P. mooniana* has an alternate arrangement of leaves and shoots, ovate to elliptical, rounded at the base of the leaf, and pointed to tapered at the tip of the leaf; the surface of the leaf is glabrous; the veins of the leaves total seven pairs; it has caducous scales; small stipules are not even present as shown in Figure 2 [24]. The oblong-shaped flowers are purple-black, and the petals are greenish [20].

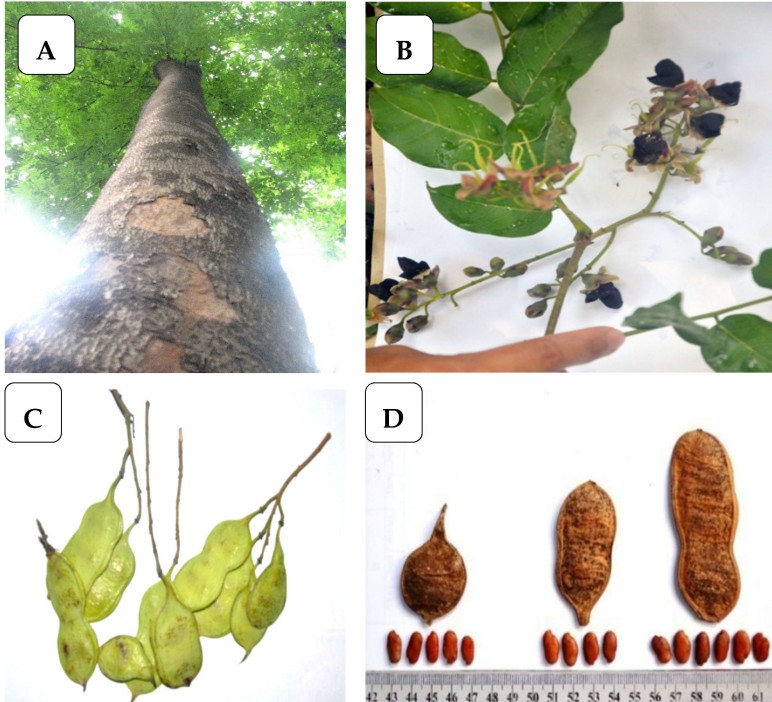

**Figure 2.** Picture of *P. mooniana*, natural stands (**A**); flowering twigs, flower parts, and pods (**B**); fruit morphology (**C**,**D**) [4,20].

*P. mooniana* has a fruiting season in September and October [6]. The fruit is classified as a boxed fruit with seed chambers or boxes of up to four chambers (generally one, two, and three seed chambers). Each chamber contains one seed, hard fruit skin (has thick cork tissue), and a single fruit or single true fruit. Young fruits are light green [20]. When ripe,

the skin splits (*dehiscence*), is inseparable from the fruit stem, and is pale white or brownish. The fruit is rounded at the base and tapered at the end [4]. The pod-shaped fruit belongs to the *indehiscent* fruit type. One pod holds 1–6 seeds, measuring 4–16 cm long and 2.5–5.0 cm wide, round at the base, and tapering at the end. The unripe pods are green and turn brownish when ripe [22]. Physiologically mature pods are brown with button-shaped seeds 1 cm in diameter and 4 mm thick [6]. The size dimension of the seed considered medium, around $0.7 \times 1.6 \times 0.5$ cm [20]. Seeds are orthodox, which means they have a hard seed coat and are difficult to germinate because they experience seed dormancy. The structure of seeds consists of *Spermodermis, Testa, Tegmen, Funiculus,* and *Nucleus semminis* [4]. Seeds of this species are classified as dicotyledonous seeds (two pieces) and have a length of 1.0–1.3 cm and a width of 0.7–0.8 cm [22].

### 3.1.2. Habitat Characteristics and Distribution

*P. mooniana* is a single species in Southeast Asia, synonymous with *Ormosia villamilii* Merr., *Pericopsis ponapensis* Hosok. The vernacular names of *P. mooniana* are nandu wood, nedun tree (En), kayu kuku (Indonesia), kayu besi papus (Sulawesi), nani laut (Irian Jaya), kayu laut (Sabah Peninsula), merbau laut (peninsula, Malaysia), and makapilite (Bisaya, Philippines) [24,25].

*Pericopsis* species were categorized as luxury wood consisting of five species, four found in the African Continent. *P. mooniana* has wide distribution covering Sri Lanka, Southeast Asia (Malaysia, Indonesia, the Philippines), and Oceania (Papua New Guinea)., The species distribution of *P. mooniana* in Indonesia includes Sumatera, Kalimantan, Sulawesi, Maluku, Halmahera, and Papua (Figure 3) [4,6,24–26].

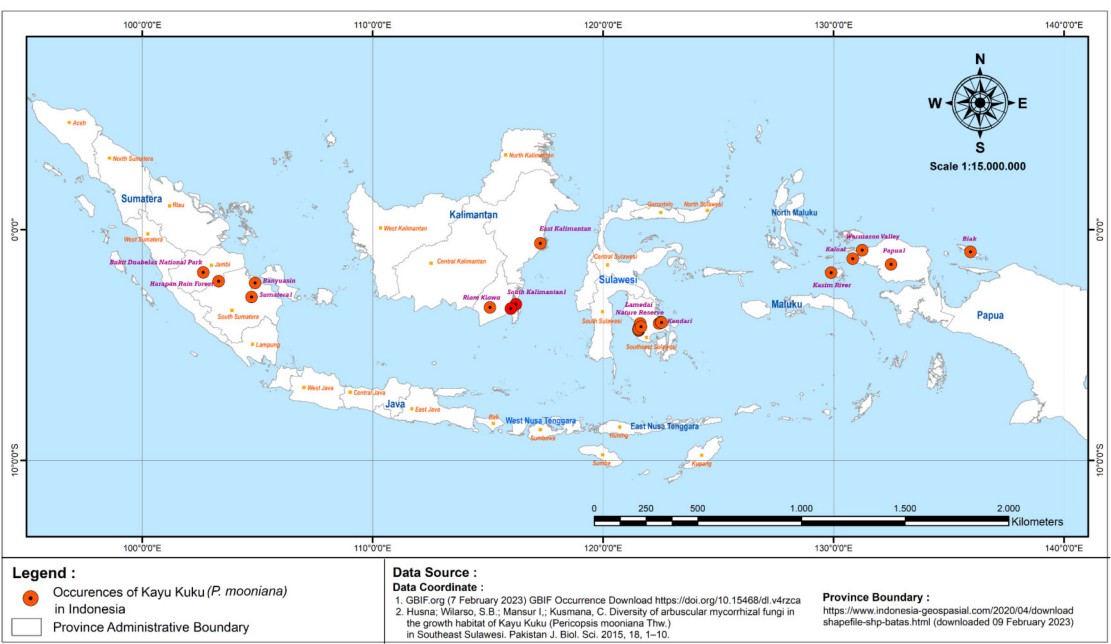

**Figure 3.** Distribution map of *P. mooniana* in Indonesia.

*P. mooniana* grows naturally in coastal, riparian, evergreen, and semi-deciduous forests [21] with relatively fertile regosol soil. This species distributed at 200–350 m asl, with annual rainfall around 750–2000 mm. This species is suitable to grow on non-stagnant soils, loamy, hilly topography with gentle slopes at an altitude of <30 m asl [4,6].

### 3.1.3. Genetic Resources of *P. mooniana*

*P. mooniana* is a rare and endangered tree species and potentially a priority species requiring further protection for evaluating, conserving, and managing forest genetic resources [8,27]. Deforestation, mining activities, over-exploitation of timber, forest conversion

to agriculture, and settlement threaten the habitat of *P. mooniana* [28]. Widyatmoko [29] reported that in 2018 more than 600 plant species in Indonesia were almost threatened with extinction. This pressure caused the population of species *P. mooniana* in their natural habitat to decline, resulting in a potential risk of extinction. Therefore, *P. mooniana* was proposed for inclusion in Annex II of the 1992 CITES convention, which requires all trade in a species to be registered (UNEP-WCMC 2014).

In addition to the exploitation and habitat disturbance, *P. mooniana* is also threatened by genetic issues such as inbreeding, loss of genetic diversity, and accumulation of gene mutations that can lead to extinction. Therefore, genetic conservation plays a vital role because it is associated with significant factors contributing to species and population extinction [30]. Genetic materials from various sources are required to conserve *P. mooniana*. Knowledge of genetic diversity is the first step in conserving species to reduce the risk of extinction by developing conservation strategies and restoration practices [31–33].

Research on the genetics of *P. mooniana* in the context of conservation in Indonesia is limited to the Lamedae Nature Reserve area of Southeast Sulawesi and the Pulau Laut of South Kalimantan [34–36]. The genetic diversity of *P. mooniana* in the four populations studied in the Lamedae Nature Reserve and surrounding populations shows moderate genetic diversity [34]. However, the genetic diversity of *P. mooniana* at the seedling level shows a reasonably high genetic diversity [35]. Meanwhile, the genetic diversity of *P. mooniana* found on the Pulau Laut of South Kalimantan shows low genetic diversity [36]. Nowadays, *P. mooniana* stands quickly disappear due to logging and land clearing. Only a few stands of *P. mooniana* remain in Papua [37], Pulau Laut South Kalimantan [38], Jambi South Sumatera [39], and Southeast Sulawesi (Lamedae Nature Reserve) [40], and this species tends to survive continuously in forest areas.

This genetic conservation activity not only keeps the number of trees and populations of *P. mooniana* but also needs for plants' development and adaptation to their environment [31,41]. Genetic conservation is one of the methods to maintain maximum genetic diversity and give opportunities for the species to adapt and evolve [31]. The results of the genetic diversity analysis of the endangered *P. mooniana* are as follows: (1) the genetic diversity ranges from low to moderate [42], (2) the distribution of genetic diversity is mainly within the population [43], (3) population distribution based on genetic distance is closely related to geographic distance [43,44] and (4) mating occurs randomly if the number of parent trees is large enough [34].

The genetic diversity of *P. mooniana* is obtained using various molecular markers, namely isozymes, RFLP (restriction fragment length polymorphism), DNA markers (deoxyribose nucleic acid), and sequencing [45–47]. In addition, it can also be used for species identification (DNA barcoding) [48], clone identification [49], and population genetic diversity [50–52]. Random amplified polymorphism DNA (RAPD) markers have been used to analyze the genetic diversity of *P. mooniana* from four populations in Southeast Sulawesi (Lamedae Nature Reserve and the villages of Lamedae, Balijaya, and Tangketada). The degree of genetic diversity is nearly identical across all populations with an average He value of =0.361 and is, therefore, classified as moderate. Lamedae Nature Reserve has the highest genetic diversity (He = 0.383 ± 0.031). The population of *P. mooniana* from the three villages has a close genetic relationship [34]. Yuskianti et al. [36] observed that the genetic diversity of *P. mooniana* in four populations on the Pulau Laut of South Kalimantan showed that the diversity was low (He = 0.191 ± 0.013). They are closely genetically related, indicating they come from a common origin.

### 3.2. Cultivation of P. mooniana

Efforts to preserve and develop *P. mooniana* trees are strongly influenced by advances in cultivation techniques. The extinction of a species is mainly due to over-exploitation that is not balanced with a proper cultivation system, thus causing a decrease in natural germplasm sources and resulting in the existence of a species becoming rare or endangered [53]. This also works for high economic value species due to their luxury wood and

being liked by the community [3,9,34]. Efforts to preserve *P. mooniana* in the context of sustainability are strongly influenced by the cultivation level and its technological developments. The scope of cultivation includes seed technology, nursery techniques, generative and vegetative propagation, and identification and control of pests and diseases that attack the *P. mooniana* trees.

3.2.1. Seed Technology

Plant cultivation begins with conceiving seed technology, starting from understanding the phenology of flowers and fruit to the right time to collect fruit [54]. Research on the flowering and fruiting period of *P. mooniana* is limited. Knowledge of the flowering period in forest plant species is essential if breeding activities are carried out by knowing the right time for fertilization or crosses [55]. *P. mooniana* fruits every year, meaning the potential for generative propagation is very high because it is not constrained in the fruiting period. However, until now, there is no information about the potential production of *P. mooniana* seed per tree, but data on the number of seeds in one pod are available [22].

The weight of 1000 seeds is 2546 g, and 1 kg contains ± 4000 seeds. The seed is classified as an orthodox seed, with the seed's initial water content being below 10% [22]. Therefore, *P. mooniana* seeds can be stored for a long time in cold storage conditions. The moisture content of the seeds for storage can be decreased below 8% without significantly reducing its viability. Even seeds of the same genus, namely *P. elata*, are predicted to be able to be stored for up to 243 years [56]. This longevity of storage indicates that the seeds of the genus *Pericopsis* can be stored for a long time while retaining their viability. It is advantageous in supporting conservation efforts. However, until now, there has been no research on the longevity storage of *P. mooniana* seeds.

Anatomically, the seed of *P. mooniana* has a reasonably hard seed coat with a waxy layer covering it; this is one of the obstacles in the germination of the seed [19]. The hard coat causes seed dormancy, which is classified as physical dormancy, and this condition is also found in seeds of the same genus, namely *P. elata* [56]. However, an attractive condition is seen in *P. angolensis*; although it is in the same genus, no dormancy happened [57]. Breaking dormancy is the first step to accelerate the germination of *P. mooniana* seeds. Seeds without breaking dormancy have a germination of about 2%; meanwhile, those breaking dormancy have an average germination of up to 60% more than the control [22]. The most effective breaking dormancy for *P. mooniana* seeds is wounding or sacrificing on the surface of the seed coat or soaking in hot water (80 °C–90 °C) for 10 min, followed by soaking in cold water for 48 h [22]. This technique can increase germination by more than 60%. Seed selection and treatment can speed up the germination rate. By soaking yellow and brown seeds in hot water to "scarify" them, the first germination time can be shortened, and the viability increases to 76% [23]. The scarification technique by broken seed skin leads to significant differences in the percentage of sprouts, germination, average days of germination, vigor index, number of leaves, and seedling height [58].

The average germination rate of *P. mooniana* seeds is 15–37 days, but if the dormancy is broken, the germination time will be faster, which is 6–8 days on average [19,22,59]. The speed and uniformity of germination affect the process of procuring seeds; this is related to the amount and time required to procure seeds for planting. Another method that can be used to estimate the potential viability of *P. mooniana* seeds is a rapid test method; this is possible because direct viability testing takes quite a long time. Rapid seed viability tests can be carried out in various ways, such as radiographic analysis, tetrazolium test, cutting test, and other methods [53].

Another significant activity in maintaining seed viability is the post-harvest seed storage process because of its orthodox seed. It uses an airtight container stored in a low-temperature room. Along with advances in science and technology, techniques for increasing seed vigor or invigoration can be carried out in various ways: through priming, gamma ray irradiation, and ultrafine bubbles [60–62]. However, until now, there has been

no research on seed invigoration of *P. mooniana* seeds, so this is an opportunity to increase the viability of *P. mooniana* seeds to preserve them.

### 3.2.2. Propagation Techniques

Seed procurement can be done by generative methods such as tillers and stumps, vegetative propagation, such as macro propagation (cuttings, air layering grafting, and budding), and micropropagation (tissue culture).

#### Generative

Cultivation of *P. mooniana* with generative propagation and procurement seeds should be obtained from tree sources over 15 years old, with a percentage of sprouts ranging from 80% [20]. The time required from the imbibition process to the release of the cotyledons is ±48 days, and it is ready to be weaned into polybags [19]. Generative propagation, including viability and germination rate, has been described in Section 3.2.1.

#### Vegetative Propagation

Several studies on vegetative propagation for this species include shoot-cutting techniques and tissue culture, especially in manufacturing seedlings from genetically identical superior clones [63–65]. The technique can also overcome the scarcity of seedlings, especially for species that do not fruit yearly. The shoot-cutting technique can be applied by adding as much as 60 ppm of the hormone IBA [64].

Material for tissue culture (in vitro techniques) must be well prepared. The material derived from the seeds affects the ability to sprout. Seeds that have been removed pericarp and stored at room temperature showed the highest shoot length compared to seed pericarp removal stored at 10 °C [63].

Seed germination and shoot propagation of *P. mooniana* have been investigated using in vitro techniques. Seeds were sowed in Murashige and Skoog (MS) medium without plant growth regulators and showed 100% germination. The MS + BA 0.75 mg/L treatment showed the best media treatment for shoot propagation, as indicated by the highest number of 2.3 shoots and 6.8 cm height on average at 12 weeks of observation. The tissue culture requires sterilized seeds [59], especially for the explant of *P. mooniana;* it is necessary to choose healthy, brightly colored, and not wrinkled seeds [65].

In multiple shoot cultures, establishment of seedling shoot tips *of P. mooniana* would be improved by using an MS medium with several doses of growth hormone BAP (benzyl amino purin) and NAA (naphthalene acetic acid) [66]. The highest level of micropropagation depends not only on the medium and selection of a suitable explant but also on the combination of growth and hormone [67].

### 3.2.3. Seedling Improvement Technique

The optimal temperature and humidity for germination of *P. mooniana* are at temperatures between 25 °C−30 °C with humidity between 60%−70%. Meanwhile, the best medium for germinating *P. mooniana* seeds is a mixture of soil and sand (*V*/*V*: 1/1) that has been sterilized. Normal *P. mooniana* germination is characterized by the appearance of two healthy, sturdy leaves. *P. mooniana* seedlings can be weaned in polybags when the seedlings look sturdy, the roots are well formed, and the seedlings are ready to wean around 3–4 weeks after sowing. The media must contain enough nutrients to support the growth of seedlings until they become ready to plant in the field [17]. *P. mooniana* are pioneer plants that grow well without shade [68]. Seedling of *P. mooniana* is maintained in the nursery for about 3–4 months until the stems are woody and the seedling medium is compact.

### 3.2.4. Pest and Disease of *P. mooniana*

Observations showed that 1-year-old *P. mooniana* seedlings in the nurseries had three pests: *Zeuzera* larva (coffee), leaf-rolling caterpillars, and grasshoppers (*Valanga* sp.). The

symptoms of the attack by larvae of *Z. coffeae* were borer on young stems, causing dry leaves and plant death. Plant death is due to the larvae burrowing the branch and moving in a vertical direction, thus damaging the xylem and phloem tissues of the plant [69]. The pest attacks were found in *P. mooniana* seeds from Tanggetada Natural Forest, with an attack of 1.65% [70]. *Z. coffeae* often attacks annual plants and tiny seedlings in forests, horticultural nurseries, and young plants, including *P. mooniana* [71].

Pests that attack young leaves of *P. mooniana* are larvae *Sylepta* sp., which causes the leaves to curl or fold, containing some larvae that are actively eating the leaves [72], and *Valanga* sp., causing leaf tearing in the leaf edge or leaf with a hole in the middle [70]. The attacks of these pests were found in the Lamedae Nature Reserve and Tanggetada Natural Forest. The leaf-folding caterpillar attack was also found on the *P. mooniana* plant in KHDTK, Malili, East Luwu, and South Sulawesi [40].

### 3.2.5. Mycorrhizal Fungi Application

Arbuscular mycorrhizal fungi (AMF) have been developed and used as a biofertilizer to support various uses, including for extensive agricultural development, forest reforestation [73], restoration of degraded land and ecosystems [74,75], land phytoremediation polluted [76], and conservation programs for endangered tropical tree species [77,78]. A total of 75 species of AMF from 23 genera and 11 families are found in Indonesia and are in symbiosis with various plants in various land use types [79]. Fifteen species of AMF from five families and nine genera have been found in the rhizosphere of *P. mooniana* in natural habitats and development areas in Southeast Sulawesi. Glomeraceae, including the dominant family and four AMF species, were first reported in Indonesia, namely *Glomus canadense*, *G. halonatum*, *Racocetra gregaria*, and *Ambispora appendicula* [80,81]. AMF isolated from the rhizosphere of *P. mooniana* has been collected, reproduced, and tested on its viability and growth both on a nursery/greenhouse and field scale (Table 1, Figures 4 and 5).

**Table 1.** Review of AMF symbiotic research with *P. mooniana*.

| AMF Species/Treatment | Media | Period (Month) | Effect | Experiment Type | References |
|---|---|---|---|---|---|
| *Septoglomus constrictum* + two days of watering | Gold tailings media | 4 | Increase shoot P levels and uptake | Greenhouse | [82] |
| *G. claroideum* and *G. coronatum* | Gold tailings media | 4 | Increase the height diameter, number of nodules, dry weight of plants, and uptake of N and P in the roots | Greenhouse | [83] |
| *G. claroideum* + field capacity 100 and 75 % | Gold tailings media | 4 | Increase plant height, diameter, total leaf area, nodulation, and dry weight | Greenhouse | [84] |
| *G. claroideum* and *G. coronatum* | Gold tailings media | 4 | Increase the uptake of N, P, Mn, and Fe leaves | Greenhouse | [85] |
| 5 g AMF inoculum Mycofer (mix. *Glomus manihotis*, *Glomus etunicatum*, *Acaulospora tuberculata*, *Gigaspora margarita*) and addition sago waste 20 g | Soil media of nickel post-mining site | 3 | Increase height, dry weight, root nodules, and absorption of K and Ca and reduce Ni by 32% | Greenhouse | [86] |
| *AMF native from rhizosfer P. mooniana* | Soil media of nickel post-mining site | 5 | Increase growth and dry weight, absorption of N, P, and K in three parts of the plants, of Ca (in stems and leaves) and of Mg in leaf tissues and root nodules; and reduce Ni content | Greenhouse | [87] |
| *C. etunicatum/Ha* | Soil media of nickel post-mining site | 5 | Increase the height, diameter, number, and area of leaves, plant dry weight, and Seedlimh quality index | Greenhouse | [88] |
| *C. etunicatum/Ha* + without treatment of Urea | Soil media of nickel post-mining site | 5 | Increase the number of leaves, nodulation, and dry weight of shoots and roots | Greenhouse | [89] |
| AMF local from rhizosfer *P. mooniana* | Overburden coal media | | Increase in the growth and dry weight of plants and also tend to increase the C, N, P, K, Ca, and Mg accumulation | Greenhouse | [90] |

**Table 1.** *Cont.*

| AMF Species/Treatment | Media | Period (Month) | Effect | Experiment Type | References |
|---|---|---|---|---|---|
| Mycofer IPB (*Glomus etunicatum*, *G. manihotis*, *Acaulospora tuberculata*, and *Gigaspora rosea*) | Ultisol | 3 | Increase growth and dry weight | Greenhouse | [91] |
| *Glomus* sp. (HA) | Ultisol | 3 | Increase height, diameter, number of leaves, dry weight, root nodules, root, and leaf length | Greenhouse | [75] |
| *Acaulospora* sp. 1 (HA) | Ultisol | 3 | Increase growth in height, diameter, number, and area of leaves | Greenhouse | [92] |
| Local AMF from *P. mooniana* rhizosphere | Inceptisol | 5 | Increase in height, diameter, number of leaves and root nodules, total dry weight, total chlorophyll, P, K, Ca, and Mg uptake | Greenhouse | [93] |
| AMF from *P. mooniana* rhizosphere from Lamedai NR (non-serpentine) and PT. Vale Indonesia (serpentine) | Nickel post-mining land | 3 | Increase survival rate, growth, dry weight, and accumulation of N, P, and K | Field | [94] |
| | | 24 | Increase height and diameter | | [95] |
| | | 36 | Increase growth in height, diameter, number, length, and width of leaves and dry weight of wet and dry leaves | | [96] |
| *G. coronatum* | Gold post-mining land | 4 | Increase the height, diameter, and dry weight of the leaves as well as the uptake of N, P, K, Mn, and Fe | Field | [83] |

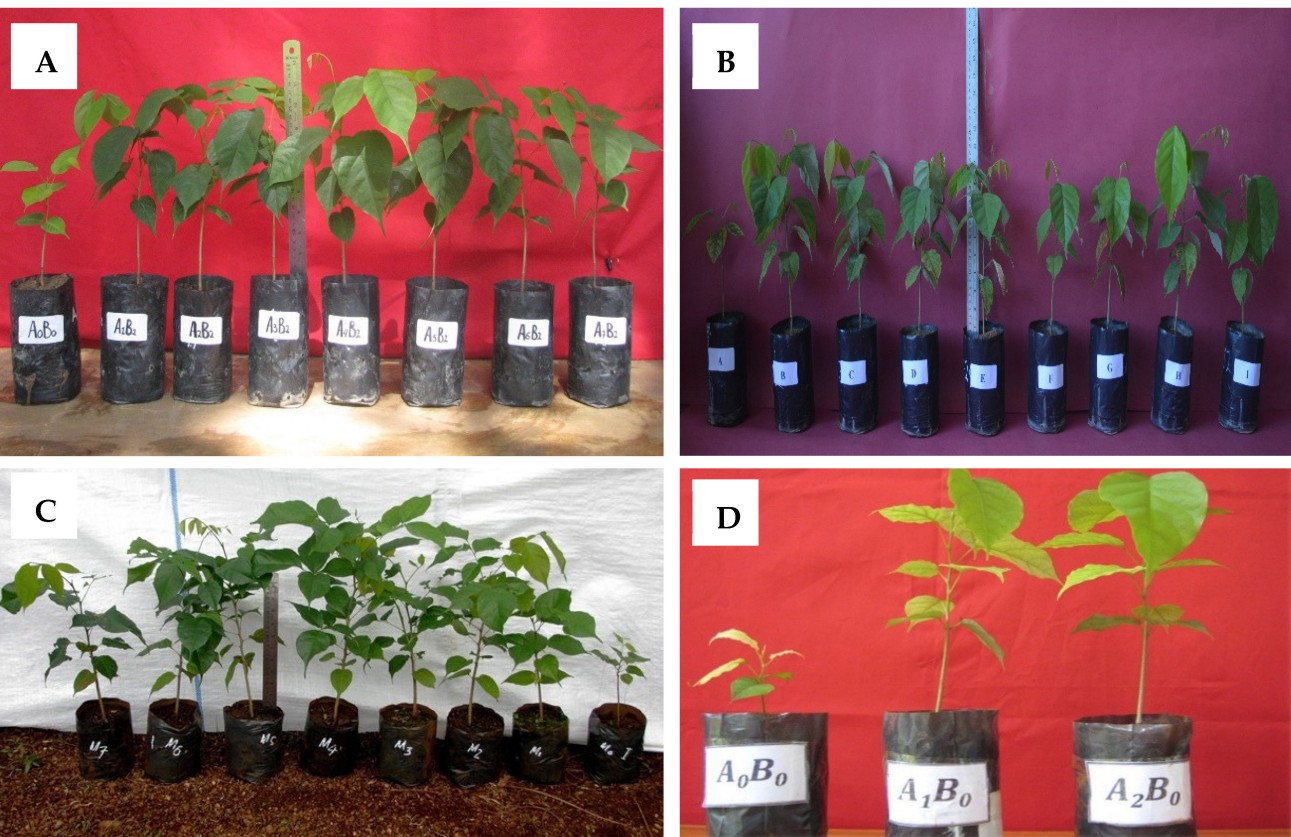

**Figure 4.** Visualization of growth performance of mycorrhizal and non-mycorrhizal *P. mooniana* at greenhouse and nursery scales. (**A**) Coal overburden (OB) media [90], (**B**) gold tailings media [83], (**C**) serpentine soil media/post-nickel mining [87], and (**D**) serpentine soil media [2].

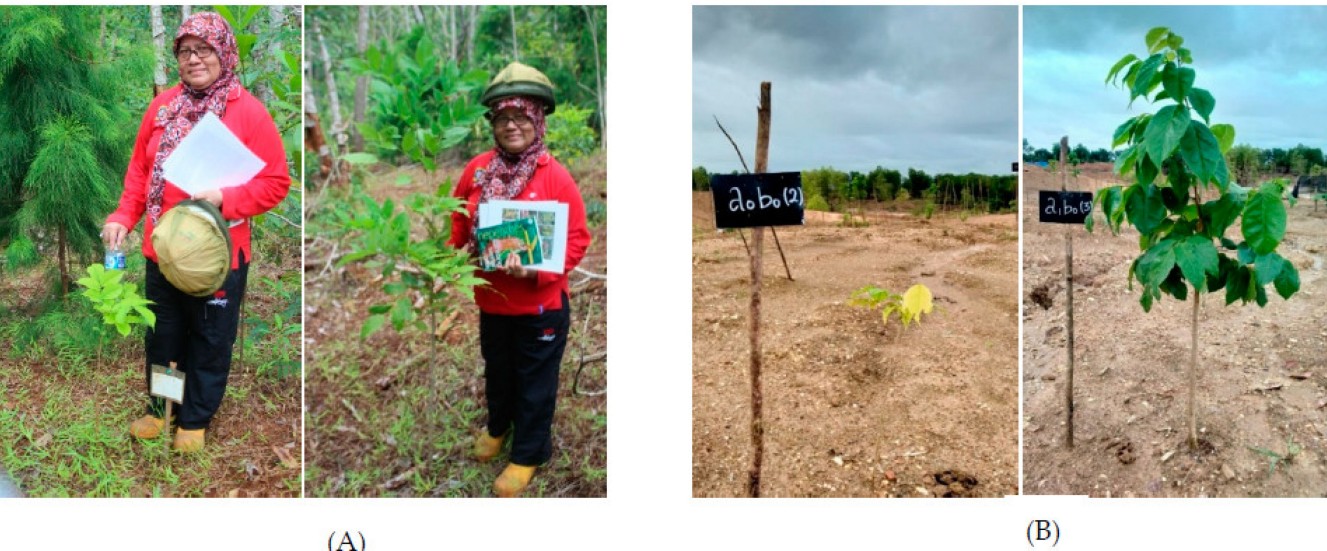

(A)

(B)

**Figure 5.** Visualization of growth performance of *P. mooniana* with and without mycorrhizal in a post-nickel field at 12 months (**A**) and a post-gold mine at four months after planting (**B**) (Photo: F.D. Tuheteru and Husna, 2022).

Table 1 shows that the application of AMF increased the growth, dry weight, nodulation, and nutrient uptake of *P. mooniana* at the nursery scale in various growing media. Based on these different research results, AMF has the potential to be developed as a biological fertilizer to support conservation programs for endangered tropical tree species and, at the same time, accelerate the success of *P. mooniana* planting in disturbed and damaged land restoration programs.

Besides AMF, *P. mooniana* also has symbiosis with rhizobium. There were 18 species of rhizobia associated with *P. mooniana* in Sri Lanka [97]. Rhizobia isolated from the roots of *P. mooniana* was effective in improving nodulation and nitrogen fixation and increasing 50% of plant dry matter under conditions of low N fertilization levels for 12 months of observation. In addition to single inoculation, double inoculation of AMF and rhizobium can increase plant growth. The synergy of AMF and rhizobium increased *P. mooniana* 2–4 fold growth compared to the control [98]. The increasing number of mycorrhizal *P. mooniana* root nodules (Figure 6) is related to the improvement in phosphorus uptake by AMF required by rhizobium [99].

3.2.6. Silviculture Techniques

Silvicultural practices such as determining plant spacing and basic and advanced fertilization greatly determine plant growth. The planting hole is made wider on post-mining land than on mineral soil. In addition, it uses the intensive application of basic and advanced fertilizers that increase the incremental growth in plant height and diameter. Wide spacing increases plant diameter. The following is a summary of publications, interviews, and observations on-site related to silvicultural practices and their effects on plant growth, namely:

1.  In the nickel post-mining area, planting holes were 60 × 60 × 60 cm. Plants were planted with a distance between planting holes of 4 m x 4 m. Before planting, the soil is mixed with compost and urea (5 kg and 100 g, respectively), one-fifth of the standard carried out by PT. Vale Indonesia Tbk and mixing sago pulp with soil media was carried out, with a height and diameter increment of 0.44 m and 0.8 cm [2,96]. *P. mooniana* was planted between reclaimed trees at a spacing of 4 × 5 m and filled with 25 kg of active compost and 0.4 kg of urea + TSP + KCl + dolomite each per planting hole. In the first and second 6 months of maintenance, 0.4 kg each of urea + TSP + KCl was added, and in the third period, pruning was conducted and

mulch was placed at the base of the stem circle. This treatment increases height and diameter increment of 0.93 m and 1.06 cm, respectively (personal communication with Mr. Guntur Sambernyowo, Retired PT Vale Indonesia).

2. In gold post-mining land, the spacing is 2 × 2 m with a planting hole of 40 × 40 × 40 cm. *P. mooniana* seedlings with a height of ±30 cm were transferred and planted in the holes with 4 kg of compost. A total of 50 g of NPK fertilizer per plant was added after one day of planting and then treated with watering and weed control. It increased height and diameter averages of 0.2 m and 0.36 cm, respectively [83].

3. In the land occupied by *Imperata cylindrica*, the planting distance was 2.5 × 2.5 m with a planting hole of 30 × 30 × 30 cm and basic fertilizer of 1 kg of compost. The mean increase in height and diameter is 0.89 m and 1.20 cm at 17 years old (observation on site).

4. In the green open space of Halu Oleo University, the planting spacing was 3 × 3 m with 30 × 30 × 30 cm planting holes and no further fertilization. The mean increase in height and diameter at 13 years old is 0.84 m and 1.75 cm (observation on site).

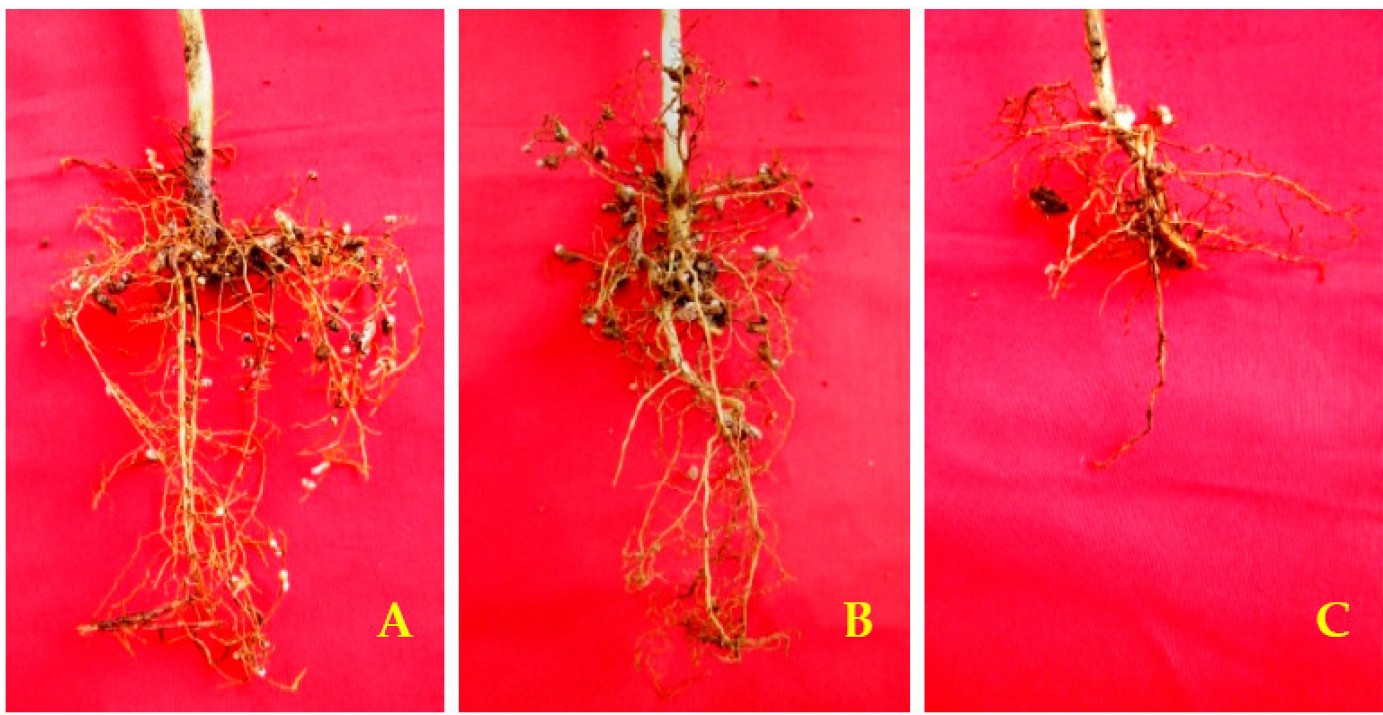

**Figure 6.** Visualization of root nodules of *P. mooniana* plant mycorrhizal (**A**,**B**) and non-mycorrhizal (**C**) [2].

### 3.2.7. Future Opportunities in Improving Seed and Seedling Quality

There is still a technological gap in *P. mooniana* cultivation in Indonesia, while natural conditions indicate that *P. mooniana* stands to decline. Therefore, it is necessary to dig more profound the technology that can fill the gap. Regarding conservation, it is essential to use quality seeds and seedlings. Suppose we identify several aspects that need to be encouraged to be developed to support conserving *P. mooniana* species. We begin with the seed source of *P. mooniana*, which until now is limited. The existing seed source is the identified seed stands and is the lowest rank in the classification of seed sources. Classification of seed sources determines the genetic quality of the plants. The genetic diversity of *P. mooniana* from several populations in Indonesia shows that originating from populations in Papua has a high level of diversity compared to other populations from Sulawesi, Kalimantan, and Java [100]. This information is essential if we develop a seed source because it will be related to the quality of the seed produced from that source.

Seed quality is also determined by post-harvest handling because it is related to the physical and physiological quality of the seed. The physiological quality of seeds can be

improved by applying invigoration technology. Until now, research on the invigoration of *P. mooniana* seeds has never been done; this condition is a challenge for forest tree seed researchers to research improving the physiological quality of *P. mooniana* seeds. Seed invigoration techniques on various forestry plant species have increased seed germination [101]. If the invigoration technique is applied to *P. mooniana* seeds, it is very promising to increase their viability when their vigor has decreased.

Regarding seed technology for conserving *P. mooniana* species, a seed coating technique can also be used. Seed coating is wrapping seeds by mixing various microbes such as plant growth-promoting bacteria, rhizobia, arbuscular mycorrhizal fungi, and Trichoderma [102]. All of them are beneficial for the growth at the seedlings level and the plants in the field. This effort can help improve the quality of seeds and seedlings and support successful planting to preserve *P. mooniana*.

### 3.2.8. Harvesting Technique

To find out the general picture of the impact of harvesting *P. mooniana* on sustainability and the environment, knowing the activities of harvesting natural forest wood is a learning process for better management. Sustainable forest management practices also implicate timber harvesting techniques that can improve productivity and wood utilization efficiency and minimize wood waste and residual stand damages. Therefore, reduced impact logging (RIL) must be implemented in *P. mooniana* harvesting, especially in Indonesia's production forests [14].

RIL is one of the harvesting techniques that could cause minimum damage to forest harvesting activity [103,104]. RIL is a low-impact timber harvesting technique through a comprehensive approach to planning, performing, monitoring, and evaluating [105]. The RIL implementation aimed to reduce the negative impact on the environment and improve harvesting efficiency by decreasing the wood harvesting waste volume and harvesting cost. Additionally, it can enhance the production cost; create a conducive growing space; improve the company income, occupational health, and safety; and become a prerequisite for sustainable forest management. By implementing RIL, it is expected that the forest damage in the timber harvesting area could be reduced.

An exemplary implementation of RIL could improve timber production and promote sustainable forests, including biodiversity [106]. Sustainable forest management is an effort of emission reduction through a sustainable innovation to provide a low impact on the environment, including RIL technique implementation and implementation of other practices such as reducing the destructive impact of harvesting activity and the implementation of Sustainable Production Forest Management (SPFM) certification [107]. Learning from various studies in natural forests shows that the application of RIL techniques can increase the efficiency of wood utilization by an average of 12% [108] and reduce residual damage by 13.62% [109,110]. From a conservation perspective, silvicultural techniques of selective logging guarantee more forest sustainability [111] because it can increase environmental stability through the presence of remaining stands resulting in new regeneration [112]. Subtropical forests revealed that new regeneration is significant in establishing vegetation, thereby contributing to biodiversity and continued timber production [113]. Therefore, we should do better to overcome the scarcity of *P.mooniana* in Indonesia, and it is necessary to carry out sustainable forest management by implementing reduced impact logging (RIL). Reduced impact logging activities include improving felling and bucking techniques to make them more efficient and adjusting felling intensity and allowable diameter limits. In addition, it is necessary to carry out reforestation and revegetation activities outside the forest area [3].

Therefore, regarding climate change and global warming issues, the effort to maintain the ecological function of the forest is to treat and protect the forest vegetation from possible damage (deforestation and degradation). After using the RIL technique, the potential for carbon storage was reduced by 2.57 tons C/ha or 9.43 tons $CO_2$-equivalent [114]. The application of RIL can reduce residual stand damage by 50%, so it can maintain

forest carbon of 12.5%–25% greater than conventional logging (CL) and produce a high-quality residual stand in the next cycle. Furthermore, implementing RIL could increase the absorption capacity of $CO_2$ from the air by 1.7 times that of CL [115], and can potentially reduce emissions by approximately 1–7 tons of $CO_2$/ha/year [116]. This may act as the basis for future planned forest management activities involving RIL, carbon, and forest certification through the hierarchy of production forest management. The implementation of both RIL and forest certification can be facilitated through the binding of carbon financial incentives [116]. The implementation of RIL and CL has not guaranteed the species' existence. This is because species arrangement in the harvesting is limited for the species group, so scarcity of specific species becomes uncontrollable.

The research showed that the increment in the growth of the stand is greater than the wood consumption rate. If the yield regulation system only considers incremental growth, there will be an excess supply of wood, which threatens to reduce the economic value of the wood itself. Forests will be converted to plantations, settlements, fields, and others. To overcome this problem, yield regulation must consider the wood consumption rate. This term has not yet been developed, but databases related to monitoring consumption rates already exist, one of which is TEINIT (Timber Product Inventory on Information Technology) [117–120]. The simulation results show that the consumption rate of wood products for house and furniture construction is lower than that of wood harvested from natural forests using the incremental approach [121]. For this reason, a harvesting system that considers the consumption rate of certain tree species will help maintain the existence of these species and increase their economic value.

### 3.3. Importance of P. mooniana for the Wood Industry

#### 3.3.1. Wood Properties

It is necessary to know the nature and characteristics of the wood, which are related to its quality the designation [122]. *P. mooniana* wood has a beautiful pattern resembling teak, as shown in Figure 7 [123,124]. The genus Pericopsis consists of four species, three from Africa and one from tropical Asia. *P. mooniana* is heavily exploited for wood trade in Southeast Asia [125]. It is categorized as the fancy wood class II in the Decree of the Minister of Forestry No. 163/Kpts-II/2003 [126]. This wood species has a beautiful pattern, so it is often used as a substitute for teak (*Tectona grandis*) to make furniture, cabinets, and other construction purposes [127,128].

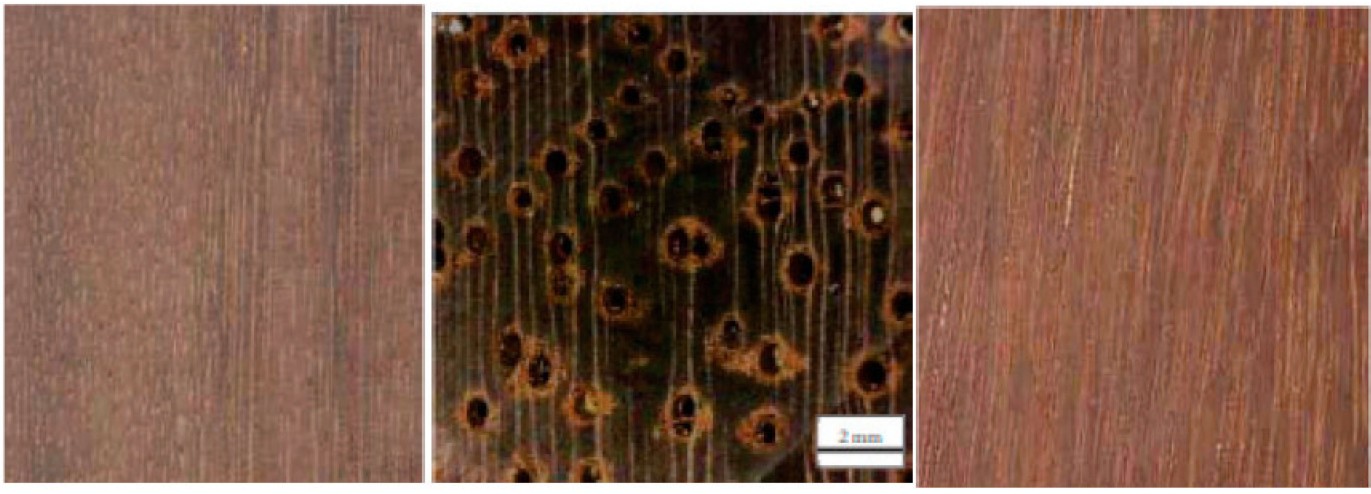

Radial section      Cross section      Tangential section

**Figure 7.** *P. mooniana* wood in macroscopic section, reproduced with permission from Krisdianto and Dewi [124].

The wood has physical properties, including specific gravity of 0.87, strong class II, radial shrinkage of 2%, and tangential shrinkage of 2%. Thus, it is classified as a low-shrinkage wood and relatively stable. Wood durability is indicated as resistant toward destructive organisms and grouped into class II. The wood's mechanical properties include a bending strength of 954 kg/cm$^2$ (wet) and 1473 kg/cm$^2$ (dry), hardness of 752 kg, maximum compressive strength of 485 kg/cm$^2$ (wet) and 699 kg/cm$^2$ (dry), and stiffness of 95,000 kg/cm$^2$ (wet) and 110,000 kg/cm$^2$ (dry). Regarding machining and workmanship, this wood has poor quality in drilling, engraving, molding, trimming, turning, and polishing [124,129]. The wood properties and characteristics are related to the wood quality. Anoop et al. [128] stated that the higher the specific gravity of the wood, the higher the fiber volume and the lower the vessel volume. The color difference between sapwood and heartwood is very conspicuous. The heartwood is brown-purple-black with light brown sapwood [130]. It differs from *P. elata* heartwood, which is brown to yellowish with streaks or without streaks, and the sapwood is distinct from the heartwood color [131].

### 3.3.2. Wood Anatomy Structure

The tangential vessel diameter of this wood is 116.7 μm and belongs to a relatively small diameter class which causes a fine texture [130]. The wood anatomy structures included vessels in a round to oval form that are diffuse-porous, arranged solitarily, and joined radially by 2–3 vessels, the number of vessels: 7–8/mm$^2$, ray height: 234–237.9 micrometers (μm), ray width: 29.6–30.4 μm, fiber length: 1142.6–1255.2 μm, fiber lumen diameter: 7.4–8.6 μm, and fiber wall thickness: 4.9–5.1 μm. The rays' cells of *P. mooniana* are arranged in an orderly manner so that they have a distinctive and attractive pattern known as the fancy wood group with a ripple mark. It has wing-shaped (aliform) to confluent parenchyma cells. Its rays' cells are uniseriate, biseriate, and multiseriate with a heterocellular composition that consists primarily of procumbent, and there are cells that are upright like a cube. The rays on the tangential section appear in a regular and distinctive arrangement [132]. This wood fiber has medium fiber lengths and thick cell walls [124,130]. Ishiguri et al. [127] stated that the wood structure is a vessel diameter of 160–170 μm, vessel frequency of 8–10 vessels/mm$^2$, fiber length of 1.0–1.7 mm, fiber cell diameter of 15–20 μm, and fiber cell wall thickness of 3.5 μm.

### 3.3.3. Wood Uses

*P. mooniana* is known as a luxury wood with a specific gravity of 0.87, strong class II, and durable class II [126], so it is suitable for indoor and outdoor furniture, window and door panels, veneer, luxury plywood, decorative, parquet flooring, bridges, and construction materials [1,124]. It has a beautiful pattern and is often used as a substitute for teak [126–128]. It is a pride and prestige for the upper middle class to use luxurious and expensive wood furniture. It offers a sense of safety and environmental friendliness and is picturesque and easy to shape. In recent years, most people have preferred wooden furniture due to its stylish interior design, accentuating aesthetic appearance, and long lifetime [133]. It can be used for a beautiful veneer with a beautiful wood pattern that is made using a slicing method to maintain its beautiful pattern and minimize incision waste by about 5% [4,134]. It is used for surface coating furniture, interior pillars, decorative walls, and a face veneer for composite wood products, including plywood, particle board, and fiberboard [135].

### 3.4. Economics and Regulation

As mentioned, *P. mooniana* wood is luxurious, fancy, and expensive wood with high economic value in the international market [2,136]. The estimated price is similar to teak wood products [136]. However, Suhartati et al. [4] pointed out that the price of exported *P. mooniana* wood is 2–3 times the price of teak wood due to its excellent wood characteristics. Since then, the wood has been commercially categorized as fancy and luxury wood similar to bongin (*Irvingia malayana* Oliv), bungur (*Lagerstroemia speciosa*), cempaka (*Michelia* spp, and cendana (*Santalum album*) as stated in Forestry Minister No. 163/Kpts-II/2003.

### 3.4.1. Trading and Its Regulations

The *Pericopsis* wood products (Figure 8) have been traded locally, so it is difficult to get quantitative data on its wood supply and demand. The wood is utilized increasingly without replanting efforts [3] and exported since 1972 and exploited uncontrollably [4], so a decrease in the population of *P. mooniana was* reported by Rain Forest Action in the year 2004 [6]. This condition has put the trees as vulnerable species (VU A1cd ver 2,3) in the IUCN Red List of Threatened Species. These species are also included as tree species in the Southeast Asia region and are immediately saved for protection from vulnerable threats by the UNEP World Conservation Monitoring Centre [9].

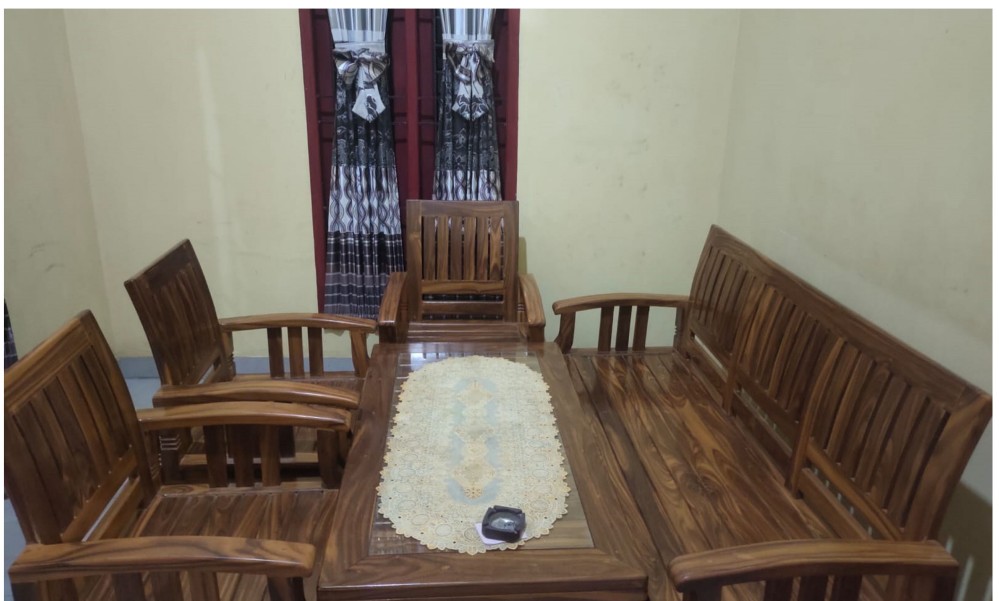

**Figure 8.** Wood product of *P. mooniana* from Southeast Sulawesi, Indonesia (Photo: F.D. Tuheteru, 2023).

The Natural Resources Conservation Institute Sulawesi Tenggara in 2012 reported that the potency of *Pericopsis* trees at the Nature Reserve (NR) of Lamedae significantly declined, was no longer dominant [3], and became vulnerable tree species [6]. In order to prevent the population decline, the Government of Indonesia, through the Ministry of Forestry, has issued its regulation number: 209/kpts- II/1994. This regulation has appointed the NR of Lamedae as a location for sustaining the *Pericopsis* stand population [6].

The history of the appointment of Lamedae NR as a location for sustaining the *Pericopsis* stands population has been well described [9,137]. It began with the letter of the Director of Nature Protection and Preserving, Ministry of Agriculture No: 1058/IV-i/IV/6/1972, dated 22 August 1972, proposing to Governor to appoint the Lamedae-Tangketada forest compartment that grows abundant *P. mooniana* trees as NR areas. Thus, the Governor agreed by issuing a recommendation letter No: PTA/4/1/11. On February 16, 1974, the Ministry of Agriculture determined the Lamedae-Tangketada forest as NR with a total area of 500 Ha. In 1987, the total area of Lamedae NR became 635,15 Ha after completing its border areas. Based on its final border areas, the Ministry of Forestry issued a Forestry Minister No. 209/KPTS-II/1994, dated April 30, 1994, for determining Lamedae forest as NR [9] and that the Lamedae NR is only one of the in situ conservation for *Pericopsis* trees in Indonesia, particularly in Southeast Sulawesi.

### 3.4.2. Sustainable Wood Supply

The vulnerability of *P. mooniana* will threaten its wood supply to wood processing industries. In order to increase the population of *P. mooniana*, the species can be chosen as the main species for rehabilitating and restoring the critical forest and land areas in its near-native habitat. There are many ways to rehabilitate critical areas, one of which is to

implement an appropriate and effective agroforestry system to restore critical land by using the agri-horti-silviculture model [138]. This model is a technique for cultivating local forest trees with certain food plant species and conserving local species more than producing commercial food [138]. This method allows local species with vulnerable categories to be conserved while combining them with appropriate annual crops.

In conservation, local tree species such as *P. mooniana* have a relatively high probability of being productively conserved with agroforestry techniques using the agri-horti-silviculture model [138]. Due to its pioneering nature and being able to grow on degraded soils, its combination with other plants in an agroforestry cropping pattern will produce at least two main outputs, namely: (1) preservation of the studied species in a productive conservation model; (2) restoration of land quality through revegetation activities.

Using *P. mooniana* as shade stands in agroforestry systems with the aim of revegetation of critical lands has an excellent opportunity to be actualized. However, few reports are available regarding *P. mooniana* agroforestry practices by local farmers [136]. They have experimented with agroforestry patterns using *P. mooniana* as shade trees and patchouli (*Pogestemon cablin* Benth.) as its intercrop. The results showed that the shade treatment significantly affected growth in stem height, the number of leaves, and the concentration of biomass of patchouli plants [136].

*3.5. Discussion and Conservation Strategies for P. mooniana*

The main obstacles to *P. mooniana* cultivation in Indonesia are (i) a decline in its natural stands and (ii) a technology gap for its cultivation. Hence, it requires several aspects to support its species conservation. The priority for species conservation needs appropriate planning in determining its conservation strategy. It begins with genetic conservation efforts that are carried out by establishing in situ and ex situ conservation areas with proper methods, planning, monitoring, and evaluation. The preparation of conservation plans fully requires information on species taxonomy, species biology, species distribution, and current population numbers in nature [139]. In this case, the genetic diversity of *P. mooniana* is a fundamental element for increasing the productivity of forest plantations and their availability in the future. In addition, genetic-based ecosystem restoration is a new strategy for genetically *P. mooniana*.

There are two genetic resource conservations: in situ and ex situ. The in situ conservation for genetic conservation can be done by (1) selecting a representative population of each natural distribution and (2) selecting populations with variations in morphological characters and high genetic diversity. It is more effective and realistic than ex situ, as well as protects the reservoir of genes for potential use in the future at conservation areas. A major problem with in situ conservation is the conflict between reserves and local people. Hence, involving local communities in conservation efforts becomes essential. Although the interests of local people and the conservation managing authorities differ, efficient management of the buffer-zone areas has become a possible solution. The next step is maintaining the number of individuals and periodically managing genetic analysis. The in situ conservation of *P. moniana* has been supported legally in Southeast Sulawesi through the Forestry Minister degree No. 209/kpts- II/1994 appointing the NR of Lamedae as a location for sustaining *Pericopsis* stand population [6].

Ex situ conservation is also an integral part of tree improvement activities. These include gene banks for seed and pollen, clone banks, breeding populations, and cryopreservation. *P. mooniana* seeds and seedlings were collected from remaining natural populations in Southeast Sulawesi and South Kalimantan and were utilized to build an ex situ conservation plot on the island of Java and provide material for the *P. mooniana* tree breeding program.

Based on the condition of *P. mooniana* in Indonesia, in situ and ex situ conservation of *P. mooniana* is a top priority to prevent genetic erosion [140]. Furthermore, coordination between the two conservation stakeholders is critical for long-term conservation sustainability through effective conservation planning [141]. The importance of in situ genetic

conservation is implemented to overcome the weakness of ex situ conservation, which cannot imitate evolutionary processes [142,143]. In situ genetic conservation is carried out by selective and adaptive processes that give rise to new genetic traits in the face of environmental challenges [144]. Furthermore, it must be based on an efficient network of protected areas and have the legal powers established by law [145]. Meanwhile, ex situ conservation is implemented as an alternative strategy when in situ conservation cannot be implemented adequately [146]. The ex situ conservation strategy of *P. mooniana* is carried out by taking and preserving samples of species, subspecies, or varieties as living plant collections in field gene banks, botanical gardens, and arboretums, or as samples of seeds, ovules, and pollen or DNA under controlled conditions [140].

Other ex situ conservation actions can be conducted through several real actions in field sites, such as preventing illegal logging, regeneration, conservation education, reforestation, agroforestry actions, and industrial forest plantation. Illegal logging can be reduced by good cooperation between the local government and law enforcers from the Ministry of Environment and Forestry [9]. Regeneration and increasing wood production for *P. mooniana* can be done by assessing the knowledge of its potency, tree distribution, wood characteristics, end-use, and type of fruits and seeds [4]. Conservation education can raise the local community's awareness of conserving *P. mooniana* by clearly explaining why this tree should be conserved. For example, it needs conservation because this tree is categorized as a vulnerable species. Its wood demand is still high due to its luxury, price, beautiful decorative pattern, and multi-used timber. Reforestation can be real action for the wide distribution of *P. mooniana* through planting in the community lands and critical forest land in Indonesia. This reforestation would assist the ex situ conservation of *Pericopsis* [9]. An agroforestry system can also be used to conserve this species. Farmers can plant seasonal crops with this species as a shading function. The research found that *P. mooniana* is reasonably correlated with other trees, so this tree is suitable as the main tree in the agroforestry system. The development of industrial plantation with *P. mooniana* as the main trees will increase the wood supply for wood processing industries.

## 4. Conclusions

The sustainability of *P. mooniana* is threatened by internal (low germination rate, limited genetic resources, and limited silviculture technique) and external (land conversion, illegal logging, and excessive logging due to its high demand) interferences. The solution for its sustainability is a strategic combination of its high demand, silviculture, and conservation effort. Its wood future demand would increase significantly due to its excellent wood characteristics. This demand should be balanced by silviculture and conservation efforts. The silviculture of *P. mooniana* has been improved through seed storage technology, improved viability and germination rate, proper micro and macro propagation, applying hormones, in vitro seed storage, and improved nursery technology and harvesting techniques. *P. mooniana* conservation can be conducted using both in situ and ex situ conservation efforts. In situ conservation is carried out by protecting its mother trees in natural conditions (i.e., Lamedae Nature Reserve) for producing good quality seeds and seedlings. Ex situ conservation is realized by planting seeds and seedlings to produce more wood through rehabilitating and restoring critical forests and lands because it can adapt to marginal land. On the other hand, it also has uses for mitigating climate change. Other actions required for supporting ex situ conservation are preventing illegal logging, regeneration, conservation education, reforestation, agroforestry systems applied in private and community lands, and industrial forest plantation.

**Author Contributions:** Each author (J.K., S. (Suhartati), H., F.D.T., D.I.D.A., M.A.L., R.U., R.P., Y., S. (Subarudi), L.A. (Lutfy Abdulah), R.S., T.K.W., S.S., A., M.W., T.K., E.T., W.I., A.S., N., L.A. (Laode Alhamd), D., and S. (Soenarno)) equally contributed as main contributors to the design and conceptualization of the manuscript, conducted the literature reviews, performed the analysis, prepared the initial draft, and revised and finalized the manuscript. All authors have read and agreed to the published version of the manuscript.

**Funding:** This research received no external funding.

**Data Availability Statement:** Not applicable.

**Conflicts of Interest:** The authors declare no conflict of interest.

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
