# Peer review of "Conserving Potential and Endangered Species of Pericopsis mooniana Thwaites in Indonesia"

_forests, doi:10.3390/f14020437_

Round 1
Reviewer 1 Report
The paper is welcome in the context of the forest sustainability. It is useful for researchers interested in the genetics and conservation of the tropical forest tree species. It is written in accordance with the topic of the journal. The paper is well documented and the references are relevant.
However, I suggest improving some aspects as follows:
The manuscript is far too long and difficult to focus the reader on the given information. It could be divided in 2 parts or compress the information. (There are many phrases that repeat the same idea). e.g Part 1- Silviculture Part 2- Wood harvesting and properties
Figure 2 -is not so evident, particularly the picture in the middle. As recommendation, you can group in a single one the figures 2,3,4, but with images of high quality (aspect of tree, bark, leaf and fruit).
Kayu kuku repeats too much, replace with studied species or synonyms, pronouns or substitute…. The idea of endangered species repeats again. Please, revise. Chapter 3.1.3. – is too long. The same idea of genetic importance is presented too many times. I suggest to compress the information, maybe in 2 pages, but indicated several citations [e.g. 31,42, 45, 46……]
Generally, the chapters 3.1, 3.2 are too long and could be summarised.
Table 2- Could be arranged to cover less pages.
Chapter 3.3.- Could be renamed (e.g. Importance of kayu kuku species for wood industry) with subchapters: 3.3.1.- Wood anatomy 3.3.2.- Wood properties 3.3.3. Uses.
I suggest to swich the figure 8 with 9. It would be more logical, in accordance with the text, to place first the macroscopic appearance before microscopy. Please write the magnification for figure 8 and 9.
Figure 10… is wood timber
Line 842- Ref 134 is missing. Occurs [34] ????
Lines-979-983- Revise, please! Some phrase starts with [25]???
Chapter 3.5 - Could be entitled Disscusion and Conservation strategies for kayu kuku species (latin name)
Reviewer 2 Report
This paper presents a review on P mooniana, an important forest resource for Indonesia. His treatment has potential is necesary more analitical point of view for a scientific review article. Undoubtedly it is important, but it would be very interesting if it showed data such as meta-analyses on growth germination studies, current and potential natural distribution maps. Current threats on maps, appropriation and cultural importance. The conclusions are very generic and could be correct for many other species. I suggest you work more on these aspects and be resubmitted.
Reviewer 3 Report
The work is of significant relevance. The data and information provided make it possible to fill gaps that exist for forest species with intensive use in the market.
Despite the relevant contribution, based on the analysis of available data for the species and the indication of gaps in research and existing technological information, which are necessary to expand the cultivation of the species and reduce pressure on natural populations, it would be very important for the authors have brought information and data on wood production and the market, factors indicated as responsible for the pressure that led to the inclusion of the species in CITES.
In this way, it is recommended the inclusion and analysis of this information allowing a more robust analysis of the danger that the extraction of wood of the species may be bringing to the conservation of the species and, even if it is possible to adopt safer measures for this conservation.
Round 2
Reviewer 1 Report
The paper improvements have been complied according to recommendations. It is now more logical and better scientifically approached.
Reviewer 2 Report
The article had a significant improvement since the last version I suggest it be considered for publication